# One-Year Impact of Scleral Lens Wear on Corneal Morphology in Keratoconus with and Without Intracorneal Ring Segment

**DOI:** 10.3390/healthcare14010131

**Published:** 2026-01-04

**Authors:** María Serramito, Ana Privado-Aroco, Gonzalo Carracedo

**Affiliations:** 1Department of Optometry and Vision, Faculty of Optics and Optometry, Complutense University of Madrid, 28037 Madrid, Spain; aprivado@ucm.es (A.P.-A.); jgcarrac@ucm.es (G.C.); 2Ocupharm Research Group, Department of Optometry and Vision, Faculty of Optics and Optometry, Complutense University of Madrid, 28037 Madrid, Spain

**Keywords:** contact lenses, cornea, corneal topography, keratoconus, visual acuity

## Abstract

**Highlights:**

**What are the main findings?**
Long-term scleral lens wear in keratoconus patients induces measurable changes in corneal morphology, including increased superior corneal thickness and region-specific alterations in anterior and posterior curvature.These structural changes differ between patients with and without intrastromal corneal ring segments (ICRS), suggesting that baseline corneal architecture influences the biomechanical response to scleral lens wear.

**What are the implications of the main findings?**
Despite corneal morphological changes, visual acuity remains stable over one year, indicating that scleral lenses provide sustained visual performance and may contribute to improving quality of life in keratoconus management.The observed changes highlight the need for the long-term monitoring of corneal parameters in scleral lens users, supporting personalized lens fitting strategies and advancing innovative approaches compared to foundational work in keratoconus care.

**Abstract:**

**Purpose:** The purpose of this study is to evaluate changes in corneal thickness and anterior and posterior corneal curvature after one year of scleral lens wear in keratoconus eyes and to determine their impact on visual performance. **Methods:** Sixty-five keratoconus subjects were divided into two groups: with intrastromal corneal ring segments (KC-ICRS) and without ICRS (KC). All participants wore 16.5 mm scleral lenses for 8 h daily over 1 year. Measurements included corneal thickness, anterior and posterior curvature, and high-contrast visual acuity assessed before and after lens wear. **Results:** Corneal thicknesses increased significantly in the superior region of the KC-ICRS group. In curvature analysis, the KC group showed inferior steepening and superior flattening, while the KC-ICRS group exhibited central and superior-nasal anterior flattening. Posterior curvature changes included central flattening and peripheral steepening. Visual acuity remained stable across all visits and groups. **Conclusions:** Long-term scleral lens wear induced measurable morphological changes, including increased superior corneal thickness and region-specific curvature alterations, which varied by ICRS presence. These changes did not compromise visual acuity, supporting scleral lenses as a safe and effective option for sustained vision correction in keratoconus. The findings highlight the importance of personalized fitting and monitoring strategies in clinical practice.

## 1. Introduction

Keratoconus is a progressive corneal disease marked by thinning and irregular astigmatism, which often results in severe visual deterioration and impacts quality of life [1]. Prevalence reports for keratoconus vary widely, from 0.2 up to 4790 per 100,000 people, and incidence estimates range between 1.5 and 25 new cases annually per 100,000 [2]. The management of keratoconus has evolved from spectacles and corneal rigid gas-permeable lenses to surgical interventions such as intrastromal corneal ring segments (ICRSs) and corneal cross-linking [2,3]. However, scleral lenses have emerged as a non-surgical alternative that provides excellent optical correction and comfort for patients with advanced stages of disease [4,5]. Despite their increasing use, the long-term biomechanical impact of scleral lens wear on keratoconic corneas remains poorly understood.

Scleral lenses constitute an effective, though still insufficiently exploited, therapeutic resource for managing corneal and ocular surface disorders [6]. A primary indication for scleral lenses (ScCL) is visual rehabilitation in individuals with irregular astigmatism, particularly those with keratoconus [7,8,9], pellucid marginal degeneration, keratoplasty [10], and refractive surgery ectasia [11,12]. These lenses are frequently prescribed for eyes severely affected by ocular surface disease for therapeutic purposes [13,14]. These contact lenses are manufactured from rigid gas-permeable materials to enhance oxygen transmission [15].

Scleral lenses are custom-designed with multiple curves forming a tear reservoir between the lens and cornea, ensuring no direct corneal contact. The tear reservoir beneath the lens enhances comfort, enabling prolonged wear with stable vision and proper centration [16]. However, prolonged wear may induce adverse effects on the ocular surface despite the high tolerance. Increased corneal thickness with contact lens wear is indicative of edema induced by hypoxic stress, and the potential effect of ScCL on corneal hypoxia and oedema has been reported [17].

Modern ScCL made of highly oxygen-permeable materials causes mild corneal edema (about 2%) in healthy young eyes [18,19]. Nevertheless, in eyes with fewer endothelial cells (cells lining the inner surface of the cornea) [20], ScCL wear may lead to epithelial and stromal edema, which could affect corneal integrity, vision, or contrast sensitivity. Modifications in visual acuity occur after an increase in corneal thickness from 4% to 6% [21].

Therefore, it is necessary to know the variations of corneal thickness and corneal radius curvature after contact lenses are worn and to observe visual changes. Several studies have shown the effects ScCL has on the ocular surface in healthy subjects with regular corneas, evaluating corneal thickness [19,22,23], anterior corneal curvature [22,24], and posterior curvature [25]. But it cannot be assumed that their results are similar and extrapolable to subjects with irregular corneas. Moreover, other authors analyzed the pachymetry [18,26] and anterior curvature [27] and posterior curvature [28] modifications after ScCL wear in irregular cornea subjects in the short term. However, these studies only evaluated short periods of lens wear, leaving a gap in understanding the cumulative impact of prolonged wear on corneal biomechanics and visual outcomes.

Variations in corneal thickness and anterior and posterior corneal curvature during ScCL use may impair the visual system, but so far, only short-term studies have been carried out. This work forms part of a continued line of research extending prior studies [27,28], with the objective of evaluating long-term ocular surface alterations and visual outcomes following one year of scleral contact lens wear in individuals with irregular corneas.

## 2. Materials and Methods

An experimental and prospective long-term investigation was conducted at the Optometry Clinic of the Faculty of Optics and Optometry of the Complutense University of Madrid (Spain). The research adhered to the Declaration of Helsinki [29], the institutional review board, and the good clinical practice guidelines. The investigation protocol was evaluated and approved by the Ethics Committee of the Clinic Hospital San Carlos (Madrid, Spain) with the code C.P.-C.I. 15/025-E. The information and informed consent forms were previously signed by all subjects. The methodology of this study was based on previous short-term studies [27,28]. This study began in 2017, and subjects were recruited, evaluated, and followed up from December 2017 to January 2020. Data were subsequently collected and analyzed in 2025.

A total of 68 patients were invited to participate in this study, of which 65 agreed to enroll. The three patients who declined participation cited scheduling incompatibilities that prevented them from attending the planned follow-up visits. A total of 65 subjects with irregular corneas, specifically keratoconus, were recruited and participated in this study. Subjects with ocular diseases and allergies were excluded. The sample could be divided into two differentiated keratoconus groups: keratoconus without intrastromal corneal ring segments (KC group) and keratoconus with ICRS (KC-ICRS group). The degree of keratoconus of the subjects was classified according to the Amsler–Krumeich scale between grades I and II, and the ring segment position of the KC-ICRS patients’ group was between 120° and 180° of arc implanted in the inferior area of the cornea. Demographic characteristics are shown in Table 1.

All eyes were fitted with ICD ScCL, manufactured by Paragon Vision Sciences (Mesa, AZ, USA). These lenses have an overall diameter of 16.5 mm. The ScCL had a spherical design and showed uniform power along the optical zone. It was selected following the manufacturer’s fitting guide, respecting a corneal clearance of 300 microns, as indicated by the guide.

All ScCLs were fitted by the same contact lens specialist. The corneal clearance was evaluated by an optical coherence tomography instrument (OCT; iVue-100, Optovue Inc., Fremont, CA, USA) and by diffuse light and optical section with biomicroscopy. After calculating the contact lens fitting, the parameters were sent to the manufacturer. More details about the ScCL are shown in Table 2.

All participants were instructed to wear the ScCL for at least eight hours per day for an entire year. None of the patients had used ScCL before the study, and those participants who were wearing other types of contact lenses at the time of recruitment required a one-month washout before the first evaluation in this study, thus avoiding possible artifacts on the ocular surface. The participants included in this study were required to attend five visits over 12 months: baseline visit; lens dispensing visit, where measurements were taken before lens insertion; 1-month visit; 6-month visit; and 12-month visit. At all follow-up visits, subjects were asked to attend after 8 h of ScCL wear.

Corneal topography and pachymetry were assessed using the Oculus Pentacam system (version 6.11r72; Oculus, Wetzlar, Germany) before and after lens removal following eight hours of wear. Three measurements were taken that had a test quality specification rated “OK”. All Pentacam measurements were performed by the same experienced examiner to minimize variability; however, inter-session repeatability was not formally assessed in this study. The measurements were performed in both eyes of each patient, but only one eye was selected randomly to analyze the results. Corneal anterior and posterior curvature and corneal thickness were evaluated. Posterior and anterior corneal curvature variations were analyzed at 8, 6, 4, and 2 mm corneal diameters and at 0°, 45°, 90°, 135°, 180°, 225°, 270°, and 315° meridians. Finally, corneal pachymetry was evaluated in the central and 4 quadrants: nasal, temporal, superior, and inferior.

High-contrast visual acuity was measured at 4 m with a logarithmic visual acuity chart ETDRS (0.00 LogMar = 20/20 Snellen) [30] with a Visionix VX-24 screen (Visionix VX-24 screen; Visionix, Luneau Technology, Chartres, France) in an optometric cabinet with a photopic luminance (85 cd/m^2^). Visual acuity was assessed monocularly at each visit.

### Statistical Analysis

G*Power 3 was used to calculate the estimated sample size for this study. The minimum sample size for each group was calculated using posterior and anterior corneal curvature as the primary variables based on a two-sided significance level of 0.05 and a β-risk of 0.20 (80% power). To detect a difference of 0.7 units, at least 19 subjects per group were required to achieve statistical significance. To account for an anticipated 20% loss to follow-up, the adjusted target was set at 24 participants per group. Ultimately, 65 subjects were enrolled in the study, with 42 in the KC group and 23 in the KC-ICRS group. Although the distribution was not perfectly balanced, it reflects the real-world prevalence of keratoconus patients with and without intracorneal ring segments. Data were analyzed using the SPSS 22.0 statistical software (SPSS. Inc., Chicago, IL, USA). The normal distribution of variables was assessed by the Shapiro–Wilk normality test. ANOVA analyzed differences in anterior and posterior topographic parameters across multiple visits for paired samples. Differences between pachymetric parameters before contact lens wear and after contact lens removal in different visits were analyzed by ANOVA with Bonferroni correction for paired samples (Bonferroni post hoc test). When comparing the groups, ANOVA for independent samples was performed. The results were expressed as mean ± standard deviation (SD). *p* < 0.05 was considered statistically significant.

## 3. Results

The sample age ranged from 18 to 67 years (mean ± SD: 39.39 ± 10.84 years). Of the initially recruited participants, a total of 49 patients completed the 12-month follow-up; 11 patients in the KC group and 5 patients in the KC-ICRS group dropped out mainly due to lack of time and commitment before the end of the study (Table 3).

Corneal thickness changes in different quadrants after ScCL wear are summarized in Table 4. In the KC group, there were no statistical differences between the different monthly visits. In the KC-ICRS group, there was a statistically significant increase (*p* < 0.05, ANOVA with Bonferroni post hoc correction test) in the superior area. No statistically significant differences were found between the groups.

The anterior and posterior corneal curvature results before and after 1 month of ScCL wear for KC subjects are represented in Figure 1 and Figure 2. In general, anterior and posterior corneal curvature showed flattening in the superior hemisphere and steepening in the inferior hemisphere. There was a statistically significant flattening in the anterior surface (*p* < 0.05, ANOVA with Bonferroni post hoc correction test) at a 2 mm radius in the 150° meridian and at 4 mm in the 45° and 135° meridians. The posterior surface showed a statistically significant flattening at 4 mm in the 45° meridian.

Six months later, the KC group presented a statistically significant steepening (*p* < 0.05, ANOVA with Bonferroni post hoc correction test) in the inferior corneal area anterior surface: at 2 mm in the 90° meridian; and at 6 mm and 8 mm in the 225°, 270°, and 315° meridians (Figure 1). Furthermore, the posterior surface showed a statistically significant steepening (*p* < 0.05, ANOVA with Bonferroni post hoc correction test) in the inferior area: at 2 mm in the 180° and 225° meridian; at 4 mm in the 180° meridian; at 6 mm in the 225° and 270° meridians; and at 8 mm in the 0°, 90°, 225°, and 270° meridians (Figure 2).

In the anterior corneal curvature, a statistically significant flattening occurred (*p* < 0.05, ANOVA with Bonferroni post hoc correction test) at 2 mm in the 90° and 135° meridians and at 4 mm in the 90° meridian. In contrast, a statistically significant steepening was observed (*p* < 0.05, ANOVA with Bonferroni post hoc correction test) at the inferior hemisphere: at 2 mm in the 0° and 315° meridians; at 4 mm in the 180° and 315° meridians; at 6 mm in the 0°, 180°, 225°, and 270° meridians; and at 8 mm in 225°, 270°, and 315° meridians.

Posterior corneal curvature showed changes in the inferior steepening and superior flattening. In the superior hemisphere, a statistically significant flattening was observed (*p* < 0.05, ANOVA with Bonferroni post hoc correction test) at 4 mm in the 90° meridian. Variation values were greater than the previous visit’s data.

After 1 month of ScCL wear, the anterior curvature in the anterior KC-ICRS group showed a significant steepening at the temporal area and flattening at the nasal area (Figure 1). A statistically significant steepening was presented (*p* < 0.05, ANOVA with Bonferroni post hoc correction test) at 2, 4, and 6 mm in the 135° meridian; at 4, 6, and 8 mm in the 180° meridian; and at 6 mm in the 225° and 270° meridian. The significant flattening (*p* < 0.05) was at the central corneal radius: at 2, 4, 6, and 8 mm in the 0° and 315° meridian; and at 4 and 6 mm in the 45° meridian.

In contrast, the posterior curvature showed a significant steepening at the central-inferior area; at the central corneal radius; at 4 and 6 mm in the 0° meridian; at 6 mm in the 135° meridian; at 2, 4, 6, and 8 mm in the 180° meridians; at 6 mm in the 225° meridian; and at 2, 4, and 6 mm at the 315° meridian (Figure 2).

After 6 months, both curvatures presented a significant steepening in the inferior area. The anterior surface showed a statistically significant flattening (*p* < 0.05, ANOVA with Bonferroni post hoc correction test) at the central corneal radius; at 2 mm in the 30° and 270° meridians; and at 6 and 8 mm in the 90° meridian. The posterior changes presented a statistically significant flattening (*p* < 0.05, ANOVA with Bonferroni post hoc correction test) at 2 mm in the 90° and 270° meridians and a statistically significant steepening (*p* < 0.05, ANOVA with Bonferroni post hoc correction test) at 6 mm in the 0°, 45°, 90°, and 225° meridians and at 8 mm in the 90° meridian.

The anterior changes in the KC-ICRS group after 12 months presented flattening in a general manner with respect to all corneal diameters. A statistically significant flattening was found (*p* < 0.05) at the central corneal radius, at 2 mm in the 90° and 270° meridians; at 4 mm in the 90° meridian; and at 6 mm in the 0° and 180° meridians. In the posterior cornea, there was a statistically significant flattening (*p* < 0.05, ANOVA with Bonferroni post hoc correction test) at 6 mm in the 180° meridian, and a statistically significant steepening was shown at 6 mm in the 0°, 45°, and 90° meridians.

Figure 3 provides a graphical representation of the differences in anterior and posterior central corneal curvature between baseline and after 1, 6, and 12 months of scleral lens wear in the KC and KC-ICRS groups. Expressing these changes in diopters allows for better clinical interpretation, as positive values indicate corneal steepening and negative values indicate flattening. This visual format was chosen to improve our understanding of the magnitude and direction of curvature modifications over time, complementing the numerical data presented in Figure 1 and Figure 2.

Figure 4 presents the log-MAR high-contrast visual acuity (HCVA) with ScCL at all appointments. For different irregular cornea groups, visual acuity was better with ScCL after a few months of wear; however, no statistically significant difference was found between visits (*p* > 0.05, ANOVA test for related samples).

## 4. Discussion

The first aim of contact lens fitting is to improve retinal image quality while compensating for refractive errors. Rigid gas-permeable materials are especially useful for corneal irregularities. Unlike other contact lenses that do not fully restore visual quality, these lenses, thanks to their materials and the thickness of the tear film they generate, are able to minimize higher-order aberrations, significantly improving vision [31,32].

ScCLs are becoming more commonly fitted to improve visual quality and also in dry eye cases [33,34]. Although the ability of ScCLs to improve vision is widely recognized, the long-term effects of these lenses on the ocular surface remain unclear. Short-term works have suggested that ScCLs may influence corneal thickness (pachymetry) [18,23,26] and anterior [26,27] and posterior curvature [28] in keratoconus patients. The present investigation is the first to report variations in the anterior curvature, corneal pachymetry, and posterior curvature after twelve months of ScCL wear in keratoconus subjects with and without ICRS.

This study did not show significant changes in the corneal thickness in the KC group after different visits of ScCL wear. This group manifested a light thinning of the central cornea. After 12 months, the reduction in thickness was 1.30%. The central pachymetry decrease was 0.15% in the KC-ICRS group after 12 months, and there was a 3.98% statistically significant increase in thickness in the superior quadrant. Those results are contrary to the other previous short-term study after 8 h of wearing ScCL [27], where the pachymetry in the superior area was decreased. It could be influenced by the time of ScCL use or by the reaction of the cornea with intrastromal ring segments and the ScCL above.

The reduction in central corneal thickness observed across all groups may be related to an increase in the osmolarity of the corneal clearance fluid during ScCL wear. This rise in osmolarity could promote fluid movement out of the corneal stroma, resulting in corneal thinning. In contrast, Carracedo et al. reported a decrease in corneal clearance tear film osmolarity after eight hours of ScCL wear [35].

The most notable pachymetric increase occurred in the superior cornea, possibly due to mechanical pressure from the upper eyelid acting on the lens and corneal surface. One possible explanation is that the lens distributes the pressure exerted by the upper eyelid across the entire limbal support zone. Alternatively, the eyelid could induce a slight downward decentration of the lens, leading to localized haptic compression in the superior peripheral cornea, which may cause local hypoxia and, consequently, corneal thickening. These hypotheses remain speculative and require further investigation in future studies to clarify the underlying mechanisms. In the published literature, no consensus has yet been reached on the variation of corneal pachymetry after ScCL wear. Some authors found an increase in the central cornea of no more than 2% [18,25,26,36], which coincides with the results of this study, while other authors found corneal thinning after wearing the lens [22]. However, De Luis Eguileor et al. showed a 2.3% corneal thickness increase after 12 months of wearing ScCL in patients with irregular corneas, but this was not statistically significant [37].

Regarding anterior corneal curvature, it was found that ScCL wear affected keratoconus corneas with and without ICRS in a different way. In the KC-ICRS group, there was central flattening across the corneal diameter and steepening in the superior and inferior regions. In the KC-ICRS group, there was flattening in a general manner with respect to all corneal diameters. Translating these flattening values (reaching 0.44 mm in the KC-ICRS group) into changes in anterior corneal power, clinical modifications of up to approximately 1 diopter (equivalent to 0.20 mm) could be observed once the ScCL is removed. However, the HCVA outcomes did not present statistical differences during the follow-up visits of wearing ScCL. Despite no changes being found in visual acuity with ScCL, it is possible that vision without ScCLs after their use was affected. That is to say that once the ScCL has been removed, the patient may complain of poor vision with their usual spectacle correction until the cornea has returned to its baseline state. Therefore, it would be advisable to perform topographies before and after wearing ScCL in order to inform patients about potential visual changes after lens removal. It should also be noted that measuring visual acuity under high-contrast and photopic conditions is a limited approach to assessing visual changes induced by scleral lens wear, as it does not fully reflect real-world visual performance.

Consistent with these findings, other studies found flattening of the anterior surface of the cornea after short-term use of ScCL in healthy eyes [22,24] and keratoconic eyes [26,27]. Soeters et al. [26] found a decrease in keratometry values after one week of ScCL use in subjects with keratoconus without specifying the quadrant where this flattening occurred. On the other hand, Serramito-Blanco et al. [27] showed that the flattening in the KC subjects was more marked in the nasal quadrant, whereas the variations were more pronounced in the inferior part in the KC-ICRS group, after eight hours of wearing the lens. These short-term results continue in the same line as the outcomes found for long-term results, which show that the corneal morphological changes produced by ScCL wear are stable in the long term and do not affect visual acuity with ScCL wear.

The position of the corneal ring segment could have been an influencing factor, as in both studies, it was implanted in the inferior corneal zone. Biomechanical properties and corneal stabilization improve after ICRS implantation [38], and mechanical properties may vary in different regions of the cornea, which could explain the differences observed between the KC-ICRS groups.

Furthermore, corneal edema could be another circumstance affecting corneal curvature. It could be interesting to attempt some modifications in reducing the corneal clearance thickness and know how curvature changes occur with other lens thicknesses to minimize hypoxia. Detailed analyses of tear film reservoir thicknesses in different corneal areas after wearing ScCL are necessary in all groups to confirm this hypothesis.

Referring to the posterior corneal curvature, minimal changes have been observed in normal corneas immediately following lens removal [22,25]. In this study, with irregular corneal subjects, it was found that ScCL wear affected keratoconus with and without ICRS. Regarding the KC group, it showed flattening and steepening in the superior and inferior areas, respectively. Moreover, the KC-ICRS group presented flattening and steepening in the inferior and superior regions, respectively. The differences between groups may have been influenced by the corneal biomechanical changes related to the corneal ring segment placement.

Posterior curvature changes likely have negligible visual impact, as the cornea and aqueous humor share similar refractive indices, minimizing optical significance. However, the physiological implications of this difference need to be studied. But, in this case, clinically significant edema did not occur during the follow-up time.

Currently, only one study on the influence of ScCL use on the posterior corneal surface of irregular cornea subjects has been found in the scientific literature. Serramito et al. [28] found similar outcomes; ScCL wear affected keratoconic corneas with flattening and steepening in the superior-nasal and temporal area, respectively; the ICRS group presented steepening in the superior area and flattening in the peripheral location. Nonetheless, short-term results cannot be extrapolated to this long-term work.

This experimental study has some limitations that should be taken into account. The KC-ICRS group had stromal rings placed in the inferior cornea; however, the methodology used was not sufficiently accurate to correlate without bias the position of the stromal rings and the changes in curvature and pachymetry in this region. Another limitation of this study is that the results cannot be extrapolated beyond the population included, which consisted exclusively of subjects with keratoconus, the most common indication for ScCLs. In addition, a specific lens design and diameter with a specific fitting approach were used, which may limit the generalization of the results to other types of lenses, diameters, or fitting strategies. Furthermore, to reinforce the comparison of results, it would have been very interesting to include a control group with healthy corneas. However, in this study, there was no control group; instead, a before-and-after comparison was made within the same group of patients with irregular corneas who actually use this type of contact lens. Another limitation of this study is that neither the reversibility of the changes observed nor the time required for the cornea to return to its initial values after discontinuing the use of ScCL lenses was evaluated. Having information on recovery time would provide a more complete picture of the physiological response to the use of scleral lenses. However, this was not possible in the present study, which highlights the need for further research in this direction. On the other hand, the corneal surface changes could slightly affect ocular refraction; however, no changes were found in HCVA. It could be interesting to conduct specific studies on the correlation between these parameters.

Finally, this study should be considered exploratory due to several methodological limitations. The sample size, the imbalance between groups, the spherical design of the lenses, and the absence of a control group limit the statistical power and generalizability of the results. Furthermore, the 12-month follow-up period does not allow for confirmation of the longer-term stability of corneal changes. Also, it should also be noted that measuring visual acuity under high-contrast and photopic conditions is a limited approach to assessing visual changes induced by scleral lens wear, as it does not fully reflect real-world visual performance. These factors highlight that the results obtained should not be extrapolated beyond the population studied and underscore the need for larger controlled studies with longer follow-up to validate these preliminary observations. Also, future research using toric or quadrant-specific haptic designs would be highly valuable to assess their potential influence on lens centration, limbal bearing, and regional corneal changes.

In summary, the scleral lenses fitted in this study significantly modified both anterior and posterior corneal curvature, with patterns of flattening and steepening that varied according to the type of irregular cornea. Corneal pachymetry revealed quadrant-specific changes, including a notable increase in the superior region, while high-contrast visual acuity remained stable across all groups and visits during the 12-month follow-up. This stability in visual performance, despite structural alterations, reinforces the role of scleral lenses as a safe and effective long-term option for visual rehabilitation in keratoconus, contributing positively to patient quality of life by maintaining functional vision without surgical intervention. However, further research is needed to clarify the underlying mechanisms driving these changes and to determine whether similar patterns occur in eyes with normal corneas or other ectatic disorders. Furthermore, exploring these effects in patients with severe ocular surface diseases, such as dry eye syndromes, in which scleral lenses are widely used for therapeutic purposes, could provide valuable information for optimizing care strategies in different clinical indications.

## 5. Conclusions

Scleral lenses that maintain a fluid reservoir without direct corneal contact induce significant changes in anterior and posterior corneal topography after 12 months of wear in keratoconus patients. These changes include region-specific flattening and steepening patterns, with the KC group showing superior flattening and inferior steepening and the KC-ICRS group exhibiting central and superior-nasal flattening related to ring segment position, as well as posterior central flattening. Corneal thicknesses increased particularly in the superior quadrant, likely influenced by eyelid pressure during blinking. These findings should be interpreted with caution given the exploratory nature of the study and its methodological limitations. Clinicians are advised to perform corneal topographies before and after scleral lens wear to inform patients about possible transient visual changes following lens removal, as this practice may have a significant impact on patient satisfaction and corneal biomechanics management.

Notably, these structural alterations did not affect visual performance, with high-contrast acuity remaining consistent over the year. This finding reinforces scleral lenses as a safe and effective long-term option (12 months) for visual rehabilitation in keratoconus, contributing positively to patient quality of life by preserving functional vision without surgical intervention. This study provides innovative evidence on long-term biomechanical responses and highlights the need for personalized fitting strategies and regular monitoring in clinical practice.

## Figures and Tables

**Figure 1 healthcare-14-00131-f001:**
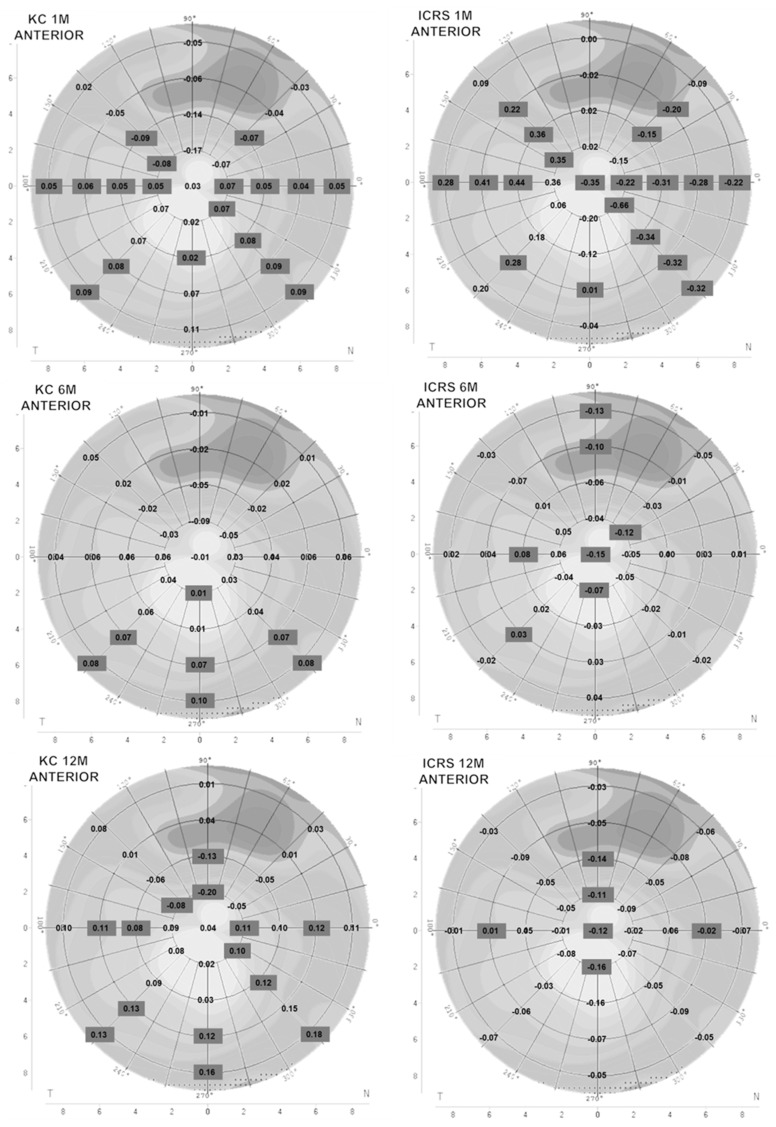
Anterior corneal curvature difference between baseline and after 1, 6, and 12 months of ScCL used in the KC and KC-ICRS groups, expressed in millimeters. Positive values expressed steepening curvature, and negative values expressed flattening curvature after use of ScCL. The values represent the mean (KC n = 31 and ICRS n = 18). Values in black squares are statistically significant. *p* value < 0.05. ANOVA with Bonferroni post hoc correction test.

**Figure 2 healthcare-14-00131-f002:**
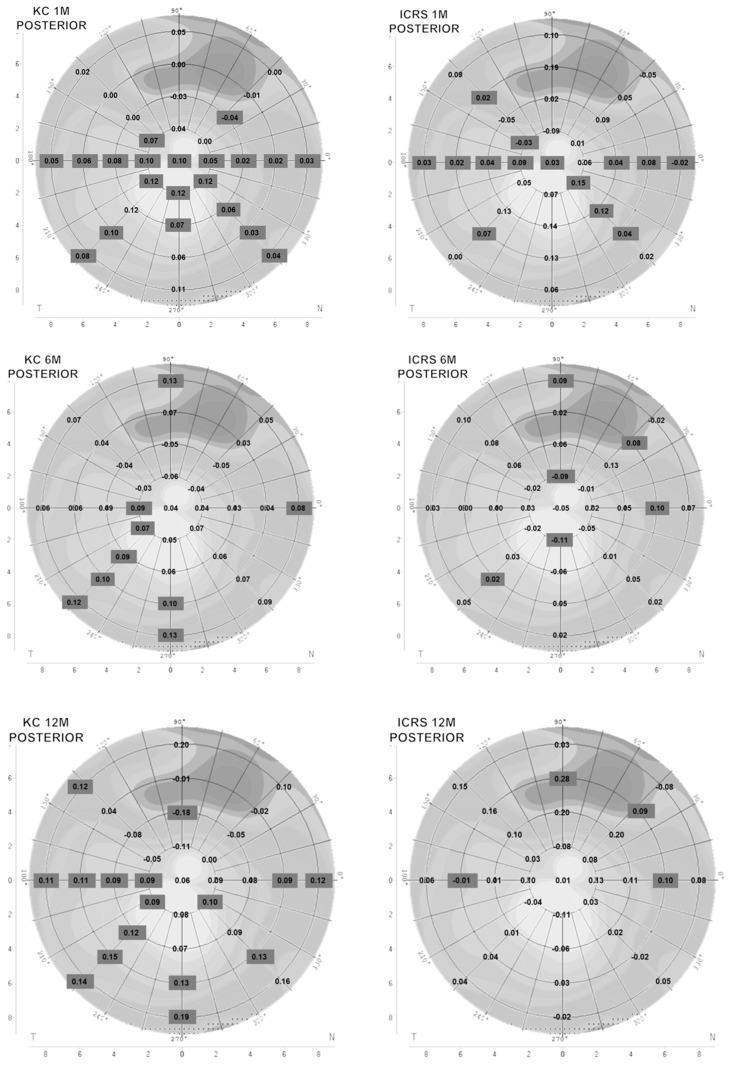
Posterior corneal curvature difference between baseline and after 1, 6, and 12 months of using ScCL in the KC and KC-ICRS groups expressed in millimeters. Positive values expressed steepening curvature, and negative values expressed flattening curvature after use of ScCL. The values represent the mean (KC n = 31 and ICRS n = 18). Values in black squares are statistically significant. *p* value < 0.05. ANOVA with Bonferroni post hoc correction test.

**Figure 3 healthcare-14-00131-f003:**
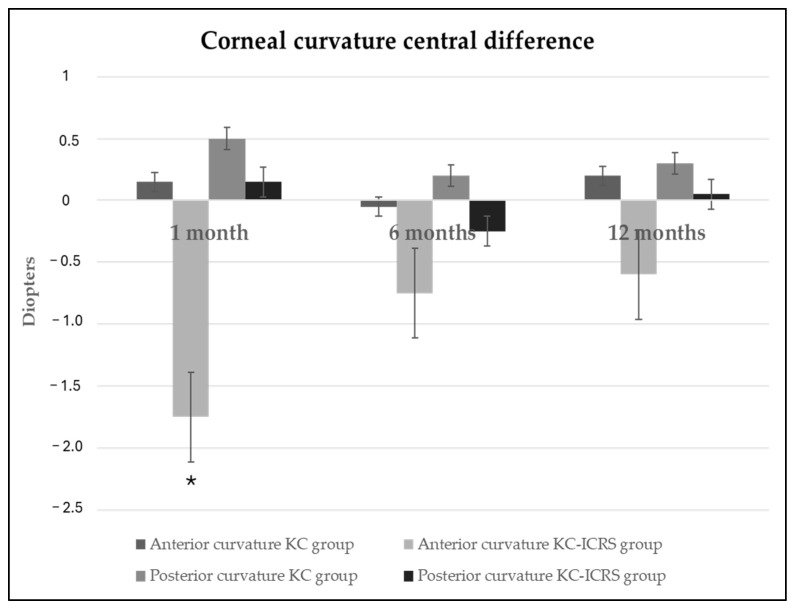
Anterior and posterior central corneal curvature difference between baseline and after 1, 6, and 12 months of ScCL wear in KC and KC-ICRS groups expressed in diopters. Positive diopters expressed steepening curvature, and negative diopters expressed flattening curvature after use of ScCL. * *p* value < 0.05. ANOVA with Bonferroni post hoc correction test.

**Figure 4 healthcare-14-00131-f004:**
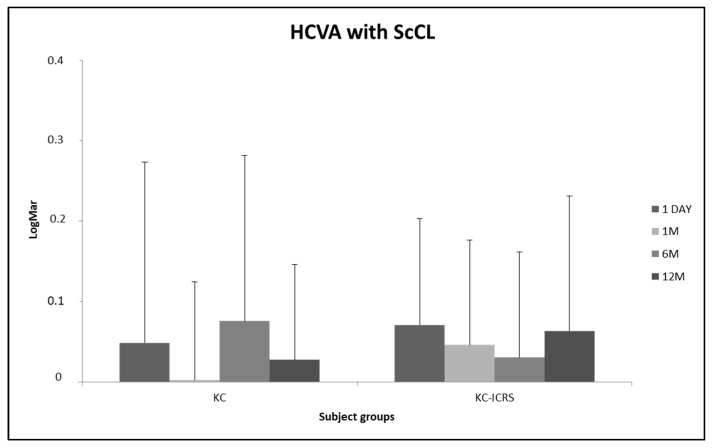
Log-MAR high-contrast visual acuity with ScCL at all appointments in different subject groups (0.00 LogMar = 20/20 Snellen). The error bars represent mean ± SD. * *p* value < 0.05. ANOVA test for related samples (KC sample n = 31; KC-ICRS sample n = 18).

**Table 1 healthcare-14-00131-t001:** Demographic characteristics of study subjects.

	Total Sample	KC Group	KC-ICRS Group
Nº subjects (n)	65	42	23
Sex (M/F)	45/20	29/13	16/7
Age (years ± SD)	39.39 ± 10.84	37.89 ± 9.19	39.64 ± 13.03
K1/K2 (D)	44.79/48.02	44.99/48.58	48.05/51.14
Mean K ± SD (D)	(46.32 ± 5.37)	(46.71 ± 3.55)	(49.53 ± 5.09)

**Table 2 healthcare-14-00131-t002:** Parameters and technical details of the contact lenses used.

Brand	ICD 16.5
Design’s owner	Paragon Vision Science
Manufacturer	Lenticon SA
Material (USAN)	Paflufocon D
Dk (barrer)	100
Water Content (%)	<1%
tc (mm)	0.30 (−3.00)
Power (D)	+1.00 D to −16.00 D
Overall Diameter (mm)	16.50
Sagittal height (microns)	3900–5600

**Table 3 healthcare-14-00131-t003:** Number of study participants at different follow-up visits.

Visits	Total Sample (n)	KC Group (n)	KC-ICRS Group (n)
Baseline visit	65	42	23
1-Month visit	57	37	20
6-Month visit	50	32	18
12-Month visit	49	31	18

**Table 4 healthcare-14-00131-t004:** Corneal pachymetry changes (± standard deviation) at baseline visit and after 1, 6, and 12 months of wearing ScCL in different groups. * *p* value < 0.05 is the ANOVA with Bonferroni correction for paired samples (Bonferroni post hoc test). Bold values indicate statistically significant changes.

Corneal Pachymetry	Visit	KC Group (n = 31)	KC-ICRS Group (n = 18)
Pachymetry apex Mean ± SD (µm)	Baseline	483.53 ± 44.77	456.40 ± 49.60
	1 Month	475.92 ± 41.80	458.72 ± 54.52
	6 Months	475.75 ± 41.34	453.00 ± 61.42
	12 Months	477.22 ± 37.93	455.72 ± 54.80
	*p*-value	0.453	0.950
Nasal pachymetry Mean ± SD (µm)	Baseline	605.60 ± 38.95	607.63 ± 42.58
	1 Month	604.15 ± 37.41	623.82 ± 48.29
	6 Months	598.51 ± 34.76	616.96 ± 46.90
	12 Months	600.44 ± 31.80	618.13 ± 48.73
	*p*-value	0.534	0.341
Temporal pachymetry Mean ± SD (µm)	Baseline	579.14 ± 40.94	566.22 ± 37.34
	1 Month	573.77 ± 38.71	573.75 ± 46.58
	6 Months	574.69 ± 38.19	574.68 ± 42.63
	12 Months	576.70 ± 38.04	566.09 ± 39.52
	*p*-value	0.778	0.988
Inferior pachymetry Mean ± SD (µm)	Baseline	580.87 ± 47.40	561.72 ± 36.20
	1 Month	577.79 ± 42.22	564.06 ± 50.28
	6 Months	572.55 ± 40.04	557.44 ± 39.39
	12 Months	574.46 ± 33.72	563.31 ± 43.96
	*p*-value	0.483	0.896
Superior pachymetry Mean ± SD (µm)	Baseline	635.07 ± 44.20	606.09 ± 57.80
	1 Month	638.45 ± 40.62	640.13 ± 59.78
	6 Months	631.82 ± 41.65	631.92 ± 70.50
	12 Motnhs	626.39 ± 37.17	630.27 ± 51.55
	*p*-value	0.398	**0.030 ***

## Data Availability

The raw data supporting the conclusions of this article will be made available by the authors upon request. The data are not publicly available due to privacy concerns and ethical restrictions.

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
