# Peer review of "One-Year Impact of Scleral Lens Wear on Corneal Morphology in Keratoconus with and Without Intracorneal Ring Segment"

_healthcare, 2026, doi:10.3390/healthcare14010131_

Round 1
Reviewer 1 Report
Comments and Suggestions for Authors
The study "One-Year Impact of Scleral Lens Wear on Corneal Morphology 2 in Keratoconus with and Without Intracorneal Ring Segment", presents itself with potentially significant clinical impact in a rapidly growing field. To provide greater transparency, it is suggested that the authors critically examine the findings more carefully, addressing the study's methodological limitations. The work has merit, but should be presented as an exploratory study.
Methods:
The reason for using visual acuity to investigate visual changes is unclear. Measuring visual acuity under high contrast and photopic illumination is a weak indicator of potential changes induced by contact lens use.
The sample size calculation should take into account losses to follow-up. Furthermore, given that more than one group will be used, there should be a reference to the minimum size of each group.
Results:
The text is overly descriptive, with many numerical indicators, making it difficult to read. A suggestion would involve a more graphic presentation of the results with a short and more focused explanation of the images and tables.
Discussion:
The authors research the minimum volunteer size to draw viable conclusions from the study. It is found in the experimental work that the number of participants who completed the study fell far short of what was proposed and with a visible imbalance between the groups. This aspect should be criticized in the work and the conclusions drawn should not be extrapolated or generalized.
Another aspect that deserves reflection in the analysis of the study's limitations is related to the follow-up time. In all evaluations, significant differences in corneal curvature were recorded, with no evidence of stabilization after 12 months.
Minor corrections:
Line 58: remove the repetition;
Line 132: State which Pentacam model was used;
Line 150: set the units for the minimum difference detected.
Reviewer 2 Report
Comments and Suggestions for Authors
Dear Authors,
I thank you for the opportunity to review your manuscript “"One-Year Impact of Scleral Lens Wear on Corneal Morphology in Keratoconus with and Without Intracorneal Ring Segment”.
I appreciate your contribution to this clinically relevant and important area that is often under-discussed - the impact of scleral lenses on corneal characteristics and biomechanics in patients with keratoconus. The study findings are important for advancing the care of patients with keratoconus in daily practice.
The abstract effectively summarizes the key findings and significance of the study.
The authors have successfully highlighted the key predictors identified in their study and the potential clinical implications.
The introduction successfully establishes the rationale for the study and concludes with a clear statement of purpose.
The authors have created a solid foundation that provides readers with the appropriate context to understand the importance of the research question since previous studies have only evaluated short periods of scleral lens wear.
The Materials and Methods section provides a thorough and transparent description of the study protocol, data collection, and analysis approach. The methodology appears appropriate for the research question and is described in sufficient detail to allow replication.
This study included two groups of participants with keratoconus - the first group had intrastromal corneal ring segments implanted, while the second group did not have intrastromal corneal ring segments.
The Results section is comprehensive and well-organized, presenting the findings in a logical sequence with appropriate statistical analysis. The authors systematically report changes in corneal thickness, as well as changes in corneal pachymetry (changes in the anterior and posterior curvature of the cornea) and further Log-MAR high-contrast visual acuity with scleral lenses in the first two groups of participants.
The Discussion section effectively interprets the study findings in the context of the existing literature on keratoconus and the implications of scleral lenses. The authors thoughtfully discuss their key findings, including that the scleral lenses implanted in this study significantly altered both anterior and posterior corneal curvature, with flattening and steepening patterns that varied depending on the type of irregular cornea, while acknowledging that scleral lenses are a safe and effective long-term option for visual rehabilitation in keratoconus, which can positively contribute to the patient’s quality of life by maintaining functional vision without surgical intervention.
They appropriately discuss the clinical implications, highlighting changes that include regionally specific corneal flattening and steepening, as well as corneal thickening in some quadrants of the cornea.
The limitations of the study are openly addressed, with constructive suggestions for future research.
The conclusion effectively synthesizes the main findings of the study and their implications.
Suggestion for improvement
The conclusion could more explicitly address the recommendation to perform topographies before and after wearing ScCL, in order to inform patients about potential visual changes after lens removal, as this has an important clinical impact on corneal biomechanics.
The reference list seems mostly comprehensive and appropriate.
A few minor suggestions for improvement:
- The authors should include a brief statement on the response rate (percentage of invited patients who agreed to participate in the study) to help readers assess the representativeness of the sample.
- Keywords - please check the MeSH database, as some of the keywords written are not listed in the MeSH database (corneal curvature; corneal thickness; scleral lens). I would also recommend alphabetical ordering of keywords.
- Line 58: "...scleral lenses scleral lenses (ScCL)..." - possible error in this sentence (please correct if necessary)
- Only 8 of the 35 listed references were published in the last 5 years. It would be advisable to add more recent and up-to-date references.
Reviewer 3 Report
Comments and Suggestions for Authors
Dear authors,
I would like to sincerely thank for the opportunity to revise your manuscript. The current study addresses an important and clinically relevant question regarding the long-term effects of scleral lens wear on corneal morphology in keratoconus patients, with and without ICRS.
The study design is appropriate to approach the question, and the main conclusions are generally supported by the presented evidence. However, several aspects should be clarified or strengthened to enhance scientific rigor and clinical relevance.
MAJOR COMMENTS
- All subjects were fitted with spherical scleral lenses despite the well-known toricity and irregularity of keratoconic corneas. The authors should justify why toric or quadrant-specific haptic designs were not considered, particularly given their potential influence on lens centration, limbal bearing, and regional corneal changes. Considering this, it is likely that morphological changes may be different.
- Although an a priori power calculation was performed, the dropout rate (65 to 49 participants) may have reduced statistical power at later visits. The authors should clarify whether the final sample retained sufficient power to detect the reported effects.
- The study reports multiple statistically significant curvature and pachymetric changes. Nevertheless, their clinical relevance is not always clearly articulated. Translating these changes (corneal curvature) into dioptric equivalents would substantially improve the applicability of the findings, enhancing the transparency of the data.
- In addition, although no visual function impairment was observed despite the curvature changes, suggesting that scleral lenses may be a safe method of visual correction, it remain unclear whether these morphological changes are clinically significant from a corneal health perspective. In other words, did the observed changes have any impact on the structural integrity or long-term health of keratoconic corneas?
MINOR COMMENTS
- The authors should clarify if the same examiner performed all Pentacam measurements and whether inter-session repeatability was assessed.
- Line 114: The authors indicate that the scleral lenses diameter were 16.5 "m". I believe that the authors wanted to say "millimeters (mm)"
Round 2
Reviewer 1 Report
Comments and Suggestions for Authors
Dear authors,
Thank you for the clarifications and additional information on the manuscript.
From my side, the work can move forward.
Reviewer 3 Report
Comments and Suggestions for Authors
Dear authors,
Thank you for taking my suggestions and comments into consideration.
I do not have any additional comments.
Congratulations on the revised manuscript.
Yours faithfully,